

# Exotic criticality in the dimerized spin-1 $XXZ$ chain with single-ion anisotropy

**Satoshi Ejima[1][*], Tomoki Yamaguchi[2], Fabian H. L. Essler[3],
Florian Lange[1], Yukinori Ohta[2] and Holger Fehske[1]**

**1** Institute of Physics, University Greifswald, 17489 Greifswald, Germany
**2** Department of Physics, Chiba University, Chiba 263-8522, Japan
**3** The Rudolf Peierls Centre for Theoretical Physics, Oxford University, Oxford OX1 3NP, UK

[*] ejima@physik.uni-greifswald.de

## Abstract

We consider the dimerized spin-1 $XXZ$ chain with single-ion anisotropy $D$. In absence of an explicit dimerization there are three phases: a large-$D$, an antiferromagnetically ordered and a Haldane phase. This phase structure persists up to a critical dimerization, above which the Haldane phase disappears. We show that for weak dimerization the phases are separated by Gaussian and Ising quantum phase transitions. One of the Ising transitions terminates in a critical point in the universality class of the dilute Ising model. We comment on the relevance of our results to experiments on quasi-one-dimensional anisotropic spin-1 quantum magnets.



# 1   Introduction

It is well established that quantum effects in one-dimensional antiferromagnetic (AFM) spin systems lead to interesting physical phenomena. While a uniform Heisenberg chain is gapless for half-integer spins, an exotic ground state with a finite gap appears for integer spins [1]. For spins $S = 1$, this Haldane phase can be understood in the framework of the Affleck-Kennedy-Lieb-Tasaki model [2, 3], whose exact ground state can be constructed in terms of valence bonds, i.e., singlet pairs of $S = 1/2$ spins. Meanwhile, the Haldane phase is recognized as a symmetry-protected topological (SPT) state [4, 5] and attracts continued attention from both theoretical and experimental points of view. For instance, the Haldane gap was confirmed experimentally in a compound with $Ni^{2+}$ ions $Ni(C_2H_8N_2)_2NO_2(ClO_4)$ [6, 7], in which a small value of the single-ion anisotropy $D$ was reported [8]. A minimal model for the description of such anisotropic spin-1 chains is

$$\hat{H}_{XXZ,D} = J \sum_j (\hat{\boldsymbol{S}}_j \cdot \hat{\boldsymbol{S}}_{j+1})_\Delta + D \sum_j (\hat{S}_j^z)^2, \tag{1}$$

where $(\hat{\boldsymbol{S}}_j \cdot \hat{\boldsymbol{S}}_{j+1})_\Delta = \hat{S}_j^x \hat{S}_{j+1}^x + \hat{S}_j^y \hat{S}_{j+1}^y + \Delta \hat{S}_j^z \hat{S}_{j+1}^z$. Assuming a positive exchange parameter $J > 0$ and $\Delta > 0$, the ground-state phase diagram exhibits three gapped phases [9]. At the isotropic point ($D = 0$ and $\Delta = 1$) the model is in a Haldane phase. A sufficiently strong single-ion anisotropy $D/J$ induces a Gaussian quantum phase transition (QPT) with central charge $c = 1$ to a topologically trivial large-$D$ (LD) phase. On the other hand, increasing $\Delta$ for fixed $D = 0$ from the isotropic point leads to a Ising QPT with $c = 1/2$ to a long-range ordered AFM phase. At larger values of $\Delta$ and $D$ there is a first order transition between the LD and AFM phases.

A natural extension of the spin-1 $XXZ$ chain (1) is the introduction of an explicit bond alternation

$$\hat{H} = \hat{H}_{XXZ,D} + J \sum_j \delta(-1)^j (\hat{\boldsymbol{S}}_j \cdot \hat{\boldsymbol{S}}_{j+1})_\Delta. \tag{2}$$

Interestingly this model realizes dimerized versions of the same three phases as the one described by Eq. (1), namely, dimerized Haldane (D-H), AFM (D-AFM) and LD (D-LD) phases. The case $D = 0$ has been studied previously [10, 11] and it was found that the D-H to D-LD transition is again of the Gaussian type, but the *entire* D-AFM-phase boundary, including the transition to the D-LD phase, belongs to the Ising universality class. A key question is how the

criticality at the phase boundary changes, if both $D$ and $\delta$ are finite. Earlier studies of half-filled Hubbard-type models realizing SPT insulating and long-range ordered (charge-density-wave) phases [12–14] indicated a transition line that is separated into continuous Ising and first-order QPTs. The meeting point of these lines belongs to the tricritical Ising universality class with $c = 7/10$, which can be described by the second minimal model of conformal field theory [15, 16].

In this paper, we determine and analyze the ground-state phase diagram of the extended model (2) by means of field theory and matrix-product-state based density-matrix renormalization group (DMRG) [17, 18] techniques, focusing on the quantum criticality at the phase boundaries. By calculating the central charge $c$, we provide compelling evidence for the existence of a critical point in the tricritical Ising universality class. Field-theory predictions for the phases and the nature of the phase boundaries of the model (2) with both single-ion anisotropy $D$ and bond alternation $\delta$ are shown to be in excellent agreement with numerical simulations. Finally, we discuss the relevance of our results to experiments on dimerized spin-1 materials [19].

## 2 Ground-state phase diagram

Let us first describe the numerical method we have used to determine the phase boundaries of the model (2). By means of the infinite DMRG (iDMRG) [20] a characteristic correlation length $\xi_\chi$ can be calculated. While this $\xi_\chi$ is always finite for fixed bond dimension $\chi$, it strongly peaks at a critical point and therefore allows for an accurate determination of QPT points, see Appendix B. This approach was already applied to half-filled Hubbard-type models [12–14].

In order to identify the different continuous phase transitions occurring in the model (2), we calculate the corresponding central charges $c$ via the entanglement entropy. For a critical system with $L$ sites and periodic boundary conditions, the von Neumann entanglement entropy of a contiguous block of $\ell$ sites with the rest of the system is $S_L(\ell) = \frac{c}{3} \ln\left[\frac{L}{\pi} \sin\left(\frac{\pi\ell}{L}\right)\right] + s_1$, where $s_1$ is a non-universal constant [21]. An accurate determination of the central charge is possible by using the relation [13, 22]

$$c^*(L) \equiv \frac{3[S_L(L/2-2) - S_L(L/2)]}{\ln\{\cos[\pi/(L/2)]\}} , \tag{3}$$

where in view of the explicit dimerization the doubled unit cell has been taken into account. Calculating the central charge numerically via Eq. (3), the universality classes of the QPT points are confirmed; this is demonstrated in Appendix B.

For iDMRG simulations typical truncation errors are $10^{-12}$, using bond dimensions $\chi$ up to 1600. In the case of finite-system DMRG calculations with periodic boundary conditions, e.g., by estimating the central charge via Eq. (3), the maximal truncation errors are about $10^{-9}$, with $\chi$ up to 6000.

Figure 1(a) shows the ground-state phase diagram of the model (2) for $\delta = 0.1$. For weak dimerization, the D-H phase survives between the D-LD and D-AFM phases. In contrast to the model without dimerization, however, the transition between the D-LD and D-AFM phases is continuous below a critical end point $(\Delta_{ce}, D_{ce}/J) \simeq (3.90, 3.64)$. Like the D-H⇌D-AFM line, this part of the transition belongs to the Ising universality class with central charge $c = 1/2$, except for the critical end point, which belongs to the universality class of the tricritical Ising model with $c = 7/10$. A tricritical Ising point at which the transition becomes first order is not observed in the dimerized model without single-ion anisotropy, simply because in this case the transition between the D-LD and D-AFM phases is always continuous. At the phase boundaries involving the Haldane phase, the universality classes are the same as in the non-

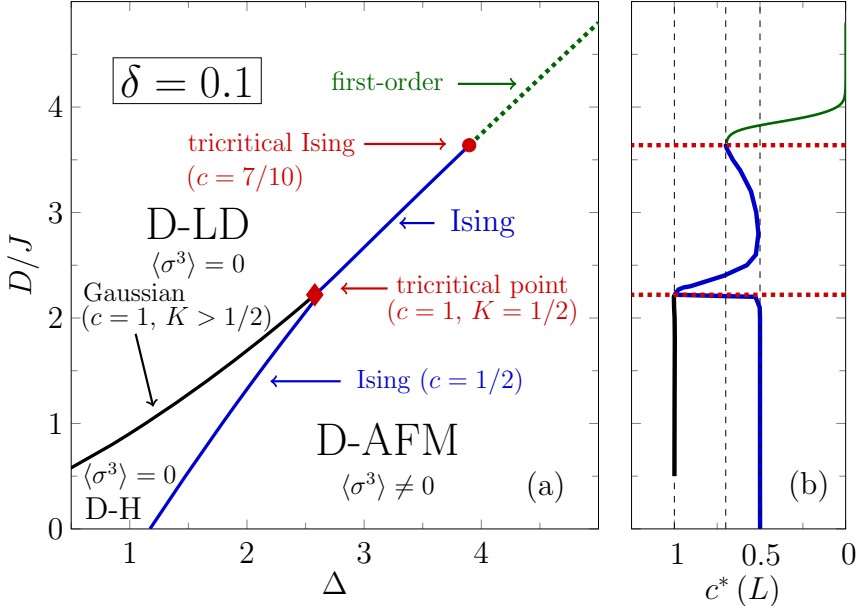

Figure 1: (a): Ground-state phase diagram of the model (2) for $\delta = 0.1$. The error bar of the tricritical (Ising) point is smaller than the symbol size. $\langle \sigma^3 \rangle$ denotes the third Ising order parameter, determining the Ising QPT between the D-H or D-LD phase and the D-AFM phase. (b): Numerically obtained central charge $c^*(L)$ on various phase transition lines from Eq. (3) with $L = 128$ and periodic boundary conditions.

dimerized model. Now the tricritical point, where the Haldane phase vanishes, is at $(\Delta_{\mathrm{tr}}, D_{\mathrm{tr}}/J) \simeq (2.58, 2.22)$. For $\delta \neq 0$, the central charge at this point is $c = 1$.

In the following, combining field theory and DMRG, we discuss various QPTs, including the direct Ising transition from the D-LD to the D-AFM phase.

## 3 Field-theory approach

In order to obtain a field-theory description of the model in the vicinity of the various phase transition lines we consider the Hamiltonian

$$\hat{H}_{\mathrm{FT}} = \hat{H} - J \sum_j (1-\alpha)(\hat{\boldsymbol{s}}_j \cdot \hat{\boldsymbol{s}}_{j+1})^2_{\Delta'}, \tag{4}$$

which differs from Eq. (2) by an additional biquadratic exchange term. A field-theory description of the model (4) can be constructed in the vicinity of the Takhtajan-Babujian point [23, 24] ($\alpha = 0$, $\delta = 0$, $D = 0$, $\Delta = 1$ and $\Delta' = 1$) following Ref. [25]. This leads to a Hamiltonian density of the form

$$\hat{\mathcal{H}} = \sum_{a=1}^{3} \frac{\mathrm{i}v_a}{2}[\hat{L}_a \partial_x \hat{L}_a - \hat{R}_a \partial_x \hat{R}_a] - \mathrm{i}m_a \hat{R}_a \hat{L}_a + \sum_{a=1}^{3} g_a \hat{J}^a \hat{J}^a + \lambda \hat{\sigma}^1 \hat{\sigma}^2 \hat{\sigma}^3, \tag{5}$$

where $\hat{L}_a$ and $\hat{R}_a$ are left and right moving Majorana fermions, $\hat{\sigma}^a$ are three Ising order parameter fields and

$$\hat{J}^a = -(\mathrm{i}/2)\epsilon^{abc}[\hat{L}_b \hat{L}_c + \hat{R}_b \hat{R}_c]. \tag{6}$$

The parameter $\lambda$ in $\hat{\mathcal{H}}$ is proportional to the dimerization $\delta$ and by virtue of the U(1) symmetry of the microscopic Hamiltonian (4) we have $v_1 = v_2$, $m_1 = m_2 \equiv m$, and $g_1 = g_2 \equiv g$.

The masses $m$ and $m_3$ are functions of $D$ and $\alpha$. The functional form of this dependence is only known in the vicinity of the Takhtajan–Babujian point and in what follows we therefore take $m_3$ and $m$ as free parameters, which we adjust in order to recover the structure of the phase diagram obtained by DMRG. Our main working assumption is that the field theory (4) remains a good description of the low-energy degrees of freedom in the vicinity of the various phase transition lines in the microscopic model even far away in parameter space from the Takhtajan–Babujian point. We note that an alternative way of deriving a field theory proposed by Schulz [26] leads to equivalent results. A third approach would be to develop a field-theory description around the SU(3) symmetric point of the spin-1 chain [27–30], but we do not pursue this here. The relation between lattice spin operators and continuum fields is

$$\hat{S}_j^a \sim \hat{M}^a(x) + (-1)^j \hat{n}^a(x) \,, \tag{7}$$

where $x = j a_0$ ($a_0$ is the lattice spacing). The smooth components of the spin operators are proportional to the currents $\hat{M}^a(x) \propto \hat{J}^a(x)$, while $\hat{n}^a(x)$ are expressed in terms of Ising order and disorder operators as

$$\hat{n}^x(x) \propto \hat{\sigma}^1(x)\hat{\mu}^2(x)\hat{\mu}^3(x), \tag{8}$$
$$\hat{n}^y(x) \propto \hat{\mu}^1(x)\hat{\sigma}^2(x)\hat{\mu}^3(x), \tag{9}$$
$$\hat{n}^z(x) \propto \hat{\mu}^1(x)\hat{\mu}^2(x)\hat{\sigma}^3(x). \tag{10}$$

In order to facilitate comparisons between field-theory and iDMRG results for the lattice model it is useful to define lattice operators

$$\hat{m}_j^\alpha = \frac{\hat{S}_j^\alpha + \hat{S}_{j+1}^\alpha}{2} \,, \quad \hat{n}_j^\alpha = (-1)^j \frac{\hat{S}_j^\alpha - \hat{S}_{j+1}^\alpha}{2} \,. \tag{11}$$

At long distances we have

$$\hat{m}_j^\alpha \approx \hat{M}^\alpha(x) \,, \quad \hat{n}_j^\alpha \approx \hat{n}^\alpha(x) \,. \tag{12}$$

It is convenient to use the U(1) symmetry to bosonize

$$\hat{L}_1 + i\hat{L}_2 \sim \frac{1}{\sqrt{\pi a_0}} e^{-i\sqrt{4\pi}\hat{\varphi}_L} \,, \quad \hat{R}_1 + i\hat{R}_2 \sim \frac{1}{\sqrt{\pi a_0}} e^{i\sqrt{4\pi}\hat{\varphi}_R} \,. \tag{13}$$

In terms of the corresponding canonical Bose field $\hat{\Phi} = \hat{\varphi}_L + \hat{\varphi}_R$ and the dual field $\hat{\Theta} = \hat{\varphi}_R - \hat{\varphi}_L$ the field theory (5) reads:

$$\hat{\mathcal{H}} = \hat{\mathcal{H}}_3 + \hat{\mathcal{H}}_B + \hat{\mathcal{H}}_{\text{int}} \,, \tag{14}$$

$$\hat{\mathcal{H}}_3 = \frac{iv_3}{2}[\hat{L}_3 \partial_x \hat{L}_3 - \hat{R}_3 \partial_x \hat{R}_3] - im_3 \hat{R}_3 \hat{L}_3 \,, \tag{15}$$

$$\hat{\mathcal{H}}_B = \frac{v}{2}\left[ \frac{1}{K}(\partial_x \hat{\Phi})^2 + K(\partial_x \hat{\Theta})^2 \right] - \frac{m}{\pi a_0} \cos\sqrt{4\pi}\hat{\Phi} \,, \tag{16}$$

$$\hat{\mathcal{H}}_{\text{int}} = \frac{2ig}{\pi a_0} \cos(\sqrt{4\pi}\hat{\Phi})\hat{L}_3 \hat{R}_3 + \lambda' \sin(\sqrt{\pi}\hat{\Phi})\hat{\sigma}^3 \,, \tag{17}$$

where $K$ is the Luttinger liquid (LL) parameter.

## 3.1 Renormalization group analysis

The most relevant perturbation is always the dimerization, and concomitantly at weak coupling the $\lambda'$ term reaches strong coupling first under the renormalization group (RG) flow. This results in a non-zero dimerization

$$\langle \hat{d} \rangle \equiv \left\langle \frac{1}{L} \sum_j \hat{D}_j \right\rangle \neq 0 \,, \quad \hat{D}_j = (-1)^j \hat{\mathbf{S}}_j \cdot \hat{\mathbf{S}}_{j+1}. \tag{18}$$

For later convenience we define a lattice version of the normal-ordered dimerization operator

$$\hat{d}_j = \frac{\hat{D}_j + \hat{D}_{j+1}}{2} - \langle \hat{d} \rangle \ . \tag{19}$$

To see what happens after the dimerization perturbation has reached strong coupling we consider the next most relevant operators, which are the Majorana mass term and the cos-term in the bosonic sector. Assuming that we have $m > 0$, what happens then depends on the sign of the Majorana mass term $m_3$. If it is positive the third Ising model is in its disordered phase $\langle \hat{\sigma}^3(x) \rangle = 0$, while $m_3 < 0$ implies that $\langle \hat{\sigma}^3(x) \rangle \neq 0$. In the latter case the strong coupling RG fixed point is amenable to a mean-field analysis. The term $\hat{\mathcal{H}}_{\text{int}}$ coupling the bosonic and fermionic sectors can be decoupled, e.g.

$$\hat{\sigma}^3(x) \sin\left(\sqrt{\pi}\Phi(x)\right) \rightarrow \langle \hat{\sigma}^3(x) \rangle \sin\left(\sqrt{\pi}\hat{\Phi}(x)\right) + \hat{\sigma}^3(x) \langle \sin\left(\sqrt{\pi}\hat{\Phi}(x)\right) \rangle \ . \tag{20}$$

This leads to a mean-field description in terms of an Ising model in a longitudinal field and a double sine-Gordon model [31, 32]

$$
\begin{aligned}
\hat{\mathcal{H}}_{\text{MF}} &= \frac{i v_3}{2}[\hat{L}_3 \partial_x \hat{L}_3 - \hat{R}_3 \partial_x \hat{R}_3] - i\tilde{m}_3 \hat{R}_3 \hat{L}_3 + h\hat{\sigma}^3 + \frac{v}{2}\left[\frac{1}{K}(\partial_x \hat{\Phi})^2 + K(\partial_x \hat{\Theta})^2\right] \\
&\quad - \frac{\tilde{m}}{\pi a_0} \cos(\sqrt{4\pi}\hat{\Phi}) + \tilde{\lambda} \sin(\sqrt{\pi}\hat{\Phi}) \ ,
\end{aligned}
\tag{21}
$$

where

$$
\begin{aligned}
\tilde{\lambda} &= \lambda'\langle \hat{\sigma}^3 \rangle \ , \quad h = \lambda'\langle \cos(\sqrt{4\pi}\hat{\Phi}) \rangle \ , \\
\tilde{m} &= m + 2ig\langle \hat{R}_3 \hat{L}_3 \rangle \ , \quad \tilde{m_3} = m_3 + \frac{2g}{\pi a_0}\langle \cos(\sqrt{4\pi}\hat{\Phi}) \rangle .
\end{aligned}
\tag{22}
$$

The classical ground state of the double sine-Gordon model is obtained by solving

$$\frac{2\tilde{m}}{\pi} \sin(\sqrt{4\pi}\hat{\Phi}_c) + \tilde{\lambda} \cos(\sqrt{\pi}\hat{\Phi}_c) = 0 \ . \tag{23}$$

Importantly, this tells us that for $\tilde{m} > 0$ we have

$$\langle \cos(\sqrt{\pi}\hat{\Phi}(x)) \rangle \neq 0 \ , \tag{24}$$

which in turn implies that

$$\langle \hat{n}^z(x) \rangle \propto \langle \hat{\sigma}^3(x) \cos(\sqrt{\pi}\hat{\Phi}) \rangle \neq 0 \ . \tag{25}$$

Hence the strong coupling RG fixed point describes a phase where antiferromagnetic order coexists with dimerization. This is the D-AFM phase identified above by the DMRG.

In the other phases the RG fixed points again occur at strong coupling but cannot be analyzed in terms of a simple mean-field argument. However, the field theory nevertheless allows for a description of the various transition lines as shown in what follows.

## 3.2 Quantum phase transitions

### 3.2.1 D-LD ⇋ D-AFM phase transition line

This corresponds to the situation where the bosonic sector remains gapped, while the third Ising model undergoes a transition between a disordered phase $\langle \hat{\sigma}^3 \rangle = 0$ on the D-LD side and an ordered phase $\langle \hat{\sigma}^3 \rangle \neq 0$ on the D-AFM side of the phase diagram. As a result the D-LD⇋D-AFM phase transition is in the universality class of the two-dimensional Ising model.

In the vicinity of the transition we may project onto the low-energy Ising degrees of freedom following e.g. Ref. [33]. Details are given in Appendix A. This yields

$$\hat{m}_j^z\Big|_{\text{low}} = A\partial_x\hat{\sigma}^3(x) + \dots\,, \tag{26}$$

$$\hat{n}_j^z\Big|_{\text{low}} = B\hat{\sigma}^3(x) + \dots\,, \tag{27}$$

$$\hat{d}_j\Big|_{\text{low}} = iC\hat{R}_3(x)\hat{L}_3(x) + \dots\,. \tag{28}$$

Along the phase transition line we thus have

$$\langle\hat{n}_j^z\hat{n}_{j+\ell}^z\rangle = B^2\ell^{-1/4} + \dots\,, \tag{29}$$

$$\langle\hat{m}_j^z\hat{n}_{j+\ell}^z\rangle = -\frac{AB}{4}\ell^{-5/4} + \dots\,, \tag{30}$$

$$\langle\hat{m}_j^z\hat{m}_{j+\ell}^z\rangle = \frac{5A^2}{16}\ell^{-9/4} + \dots\,, \tag{31}$$

and

$$\langle\hat{d}_j\hat{d}_{j+\ell}\rangle = C^2\ell^{-2} + \dots\,. \tag{32}$$

The predictions (29)–(32) are compared to iDMRG simulations below.

### 3.2.2   D-H ⇌ D-AFM phase transition line

The D-AFM to D-H transition is described by the same scenario as discussed above, since it also belongs to the Ising universality class with $c = 1/2$. Accordingly, Eqs. (29)–(32) are valid on this transition line as well.

### 3.2.3   D-H ⇌ D-LD phase transition line

As we cross from the D-AFM into the D-H phase at fixed $\Delta$ by increasing $D$ the (effective) Majorana mass $m_3$ increases. Assuming that this relation continues to hold, the characteristic energy scale in the Majorana sector can eventually become large compared to that of the bosonic sector and it is then justified to integrate out the Majorana sector. This leads to an effective low-energy description in terms of a sine-Gordon model

$$\hat{\mathcal{H}}_{\text{low}} = \frac{v}{2}\left[\frac{1}{K}(\partial_x\hat{\Phi})^2 + K(\partial_x\hat{\Theta})^2\right] - \frac{m^*}{\pi a_0}\cos(\sqrt{4\pi}\hat{\Phi})\,. \tag{33}$$

The main effect of integrating out the Majorana sector is the renormalization of the sine-Gordon coupling. Importantly, $m^*$ can vanish for particular values of $D$, which corresponds to a phase transition line described by a LL characterized by the LL parameter $K$. The low-energy projections of the lattice spin operators along this line are

$$\hat{d}_j\Big|_{\text{low}} = A_D\cos\big(\sqrt{4\pi}\hat{\Phi}(x)\big) + \dots\,, \tag{34}$$

$$\hat{n}_j^z\Big|_{\text{low}} = A_z\sin\big(\sqrt{4\pi}\hat{\Phi}(x)\big) + \dots\,, \tag{35}$$

$$\hat{n}_j^x\Big|_{\text{low}} = A_x\cos\big(\sqrt{\pi}\hat{\Theta}(x)\big) + \dots\,, \tag{36}$$

$$\big(S_j^+\big)^2\Big|_{\text{low}} = A_2\,e^{i\sqrt{4\pi}\hat{\Theta}(x)} + \dots\,, \tag{37}$$

$$\hat{m}_j^x\Big|_{\text{low}} = \frac{a_0}{\sqrt{\pi}}\partial_x\hat{\Phi}(x) + \dots\,. \tag{38}$$

This gives the following field-theory predictions for power-law decays of two-point functions

$$\langle \hat{n}_j^z \hat{n}_{j+\ell}^z \rangle = \frac{A_z^2}{2} \ell^{-2K} + \dots, \tag{39}$$

$$\langle \hat{n}_j^\alpha \hat{n}_{j+\ell}^\alpha \rangle = \frac{A_x^2}{2} \ell^{-1/2K} + \dots, \quad \alpha = x, y, \tag{40}$$

$$\langle (\hat{S}_j^+)^2 (\hat{S}_{j+\ell}^-)^2 \rangle = A_2^2 \ell^{-2/K} + \dots, \tag{41}$$

$$\langle \hat{m}_j^z \hat{m}_{j+\ell}^z \rangle = \frac{K}{2\pi^2} \ell^{-2} + \dots, \tag{42}$$

$$\langle \hat{d}_j \hat{d}_{j+\ell} \rangle = \frac{A_D^2}{2} \ell^{-2K} + \dots. \tag{43}$$

## 4 DMRG analysis

In this section, we examine various two-point correlation functions of the lattice Hamiltonian (2) using iDMRG in order to prove the field-theory predictions described in the last section. Then, the topological properties of each phase are discussed by simulating topological order parameters.

### 4.1 Quantum phase transitions

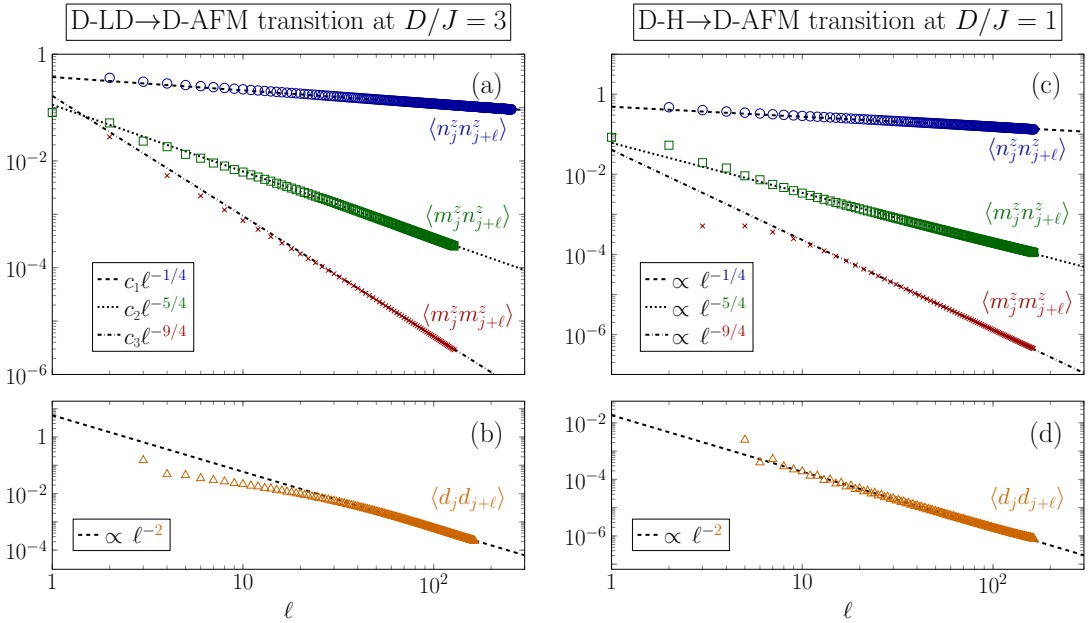

Figure 2: The connected longitudinal spin-spin (upper panels) and dimerization (lower panels) two-point functions at the Ising transition point for fixed $D/J = 3$ (left panels) and $D/J = 1$ (right panels) with $\delta = 0.1$, obtained by iDMRG with $\chi = 1600$. Correlation functions (symbols) show a power-law decay in accordance with the field-theory predictions Eqs. (29)–(32) [lines].

#### 4.1.1 D-LD ⇌ D-AFM and D-H ⇌ D-AFM Ising phase transition lines

For fixed $D/J = 3$ and $\delta = 0.1$ the Ising QPT occurs at $\Delta_c \simeq 3.303$ between D-LD and D-AFM phases as extracted from correlation length $\xi_\chi$. At this transition point various two-point

functions can be computed by iDMRG. Here, $\chi = 1600$. As shown in Fig. 2(a) field-theory predictions for diverse two-point functions of $z$-component spin operators (29)–(31) can be proved by iDMRG. Figure 2(b) demonstrates that also the dimer-dimer correlation function is in agreement with the power-law behavior according to Eq. (32) for large distances $\ell \gg 1$.

The relations between the coefficients in Eqs. (29)–(32) can be verified by fitting the iDMRG data to the field-theory predictions. For instance, in the case of the D-LD⇋D-AFM transition at $D/J = 3$ [Fig. 2(a)], we obtain $c_1 \simeq 0.381$ ($B \simeq 0.617$) and $c_3 \simeq 0.158$ ($A \simeq 0.711$), i.e., $AB/4 \simeq 0.110$, which is in good agreement with $c_2 \simeq 0.114$ from Eq. (30).

Along the Ising critical line separating the D-H and D-AFM phases the long-distance behavior of these correlation functions determined by iDMRG is again in excellent agreement with field-theory predictions, *cf.* Eqs. (29)-(32). A representative example is shown in Figs. 2(c) and (d) for $D/J = 1$ and $\Delta_c \simeq 1.789$.

### 4.1.2 D-H ⇋ D-LD phase transition line

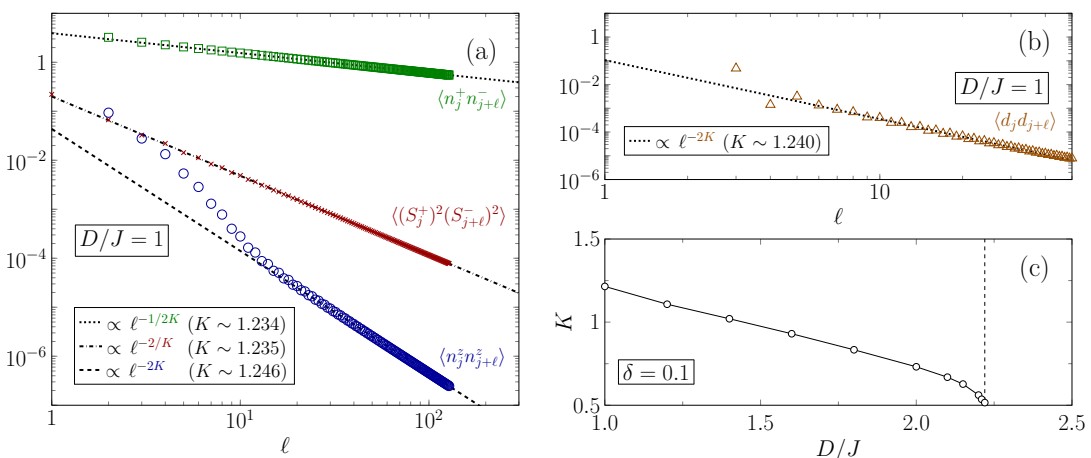

Figure 3: Spin-spin (a) and dimer-dimer (b) correlation functions at the $c = 1$ transition for $D/J = 1$ and $\delta = 0.1$ computed by iDMRG with bond dimension $\chi = 1600$. The extracted values of the LL parameter $K$ are in good agreement. (c) Extrapolated values of LL parameters $K$ via $S(q)$ of Eq. (44) on the $c = 1$ transition line for $\delta = 0.1$, obtained by DMRG with up to $L = 1024$ sites and open boundary conditions.

Along the line of Gaussian QPTs separating the D-H and D-LD phases the exponents characterizing the long-distance behavior of correlation functions depends on the LL parameter $K$ as described in Eqs. (39)-(41) and (43). In order to facilitate a comparison to the field-theory results we therefore require the LL parameter $K$. For fixed $D/J = 1$ the Gaussian transition occurs at $\Delta_c \simeq 1.135$. In Figs. 3(a) and (b) we show numerical results of correlation functions obtained by iDMRG. The values of LL parameters extracted from the fits to Eqs. (39)-(41) and (43) show reasonable agreement with each other.

These values can also be extracted from the long-distance behavior of the smooth part of the spin-spin correlation function (42), that is, the LL parameter determines the amplitude of the correlation function but not the exponent. We calculate the longitudinal spin correlation function and isolate the smooth component from a Fourier transformed structure factor

$$S(q) = \frac{1}{L} \sum_{j\ell} e^{iq(j-\ell)} \left( \left\langle \hat{S}_j^z \hat{S}_\ell^z \right\rangle - \left\langle \hat{S}_j^z \right\rangle \left\langle \hat{S}_\ell^z \right\rangle \right) \tag{44}$$

for $q \approx 0$, where $q = 2\pi/L$. The LL parameter is determined as $K = \lim_{q\to 0} \pi S(q)/q$ [34]. Figure 3(c) shows the results for the Luttinger parameter $K$ on the $c = 1$ line for $\delta = 0.1$.

At $\Delta = 1$ we have $K = 1.215$, in reasonable agreement with the values obtained from the exponents of correlation functions in Figs. 3(a) and (b). Following the Gaussian transition line by increasing $\Delta$ and $D/J$ the Luttinger parameter decreases and takes the value $K \simeq 1/2$ at the point when the Gaussian line merges with the line of Ising QPTs.

## 4.2 Topological order parameters

Let us now explore the topological properties of the phases of the model (2). Following Vidal [35], we use the infinite matrix-product-state representation formed by $\chi \times \chi$ matrices $\Gamma_\sigma$ and a positive real, diagonal matrix $\Lambda$:

$$|\psi\rangle = \sum_{\cdots \sigma_j, \sigma_{j+1} \cdots} \cdots \Lambda \Gamma_{\sigma_j} \Lambda \Gamma_{\sigma_{j+1}} \cdots |\cdots \sigma_j, \sigma_{j+1}, \cdots \rangle, \tag{45}$$

where the index $\sigma$ labels the basis states of the local Hilbert spaces. The $\Gamma_\sigma$ and $\Lambda$ are assumed to be in the canonical form:

$$\sum_\sigma \Gamma_\sigma \Lambda^2 \Gamma_\sigma^\dagger = \mathbb{1} = \sum_\sigma \Gamma_\sigma^\dagger \Lambda^2 \Gamma_\sigma. \tag{46}$$

If $|\psi\rangle$ is invariant under an internal symmetry represented by a unitary matrix $\Sigma_{\sigma\sigma'}$, then the transformed $\Gamma_\sigma$ matrices satisfy [5, 36]

$$\sum_{\sigma'} \Sigma_{\sigma\sigma'} \Gamma_{\sigma'} = e^{i\theta} U^\dagger \Gamma_\sigma U. \tag{47}$$

Here $U$ is a unitary matrix that commutes with $\Lambda$, and $e^{i\theta}$ is a phase factor. In the case of time reversal symmetry (inversion symmetry), $\Gamma_\sigma$ on the left-hand side is replaced by its complex conjugate $\Gamma_\sigma^\dagger$ (its transpose $\Gamma_\sigma^T$). Exploiting the properties of the matrices $U$ each SPT phase can be classified [5]: In the case of time reversal (inverse) symmetry the matrices satisfy $U_\mathcal{T} U_\mathcal{T}^* = \pm \mathbb{1}$ ($U_\mathcal{I} U_\mathcal{I}^* = \pm \mathbb{1}$), and the sign can be used to distinguish different SPT phases. In presence of a $\mathbb{Z}_2 \times \mathbb{Z}_2$ symmetry the order parameter is given by

$$O_{\mathbb{Z}_2 \times \mathbb{Z}_2} = \frac{1}{\chi} \text{Tr}\left( U_x U_z U_x^\dagger U_z^\dagger \right), \tag{48}$$

where we use the symmetry operations $\hat{R}^x = \exp(i\pi \sum_j \hat{S}_j^x)$ and $\hat{R}^z = \exp(i\pi \sum_j \hat{S}_j^z)$ to calculate $U_x$ and $U_z$.

In the presence of dimerization the unit cell consists of two sites, which we have to block together in order to apply the above description. For the model (2), blocking sites across weak bonds gives the same values of the order parameters as blocking across strong bonds. Figure 4 shows the iDMRG results for the order parameters in case of inverse and $\mathbb{Z}_2 \times \mathbb{Z}_2$ symmetries. If $U_x$ and $U_z$ commute ($O_{\mathbb{Z}_2 \times \mathbb{Z}_2} = 1$), the system is in a trivial phase, i.e., a site-factorizable LD state, whereas if they anticommute ($O_{\mathbb{Z}_2 \times \mathbb{Z}_2} = -1$), the system realizes a non-trivial Haldane state. If the symmetry is broken, we set $O_{\mathbb{Z}_2 \times \mathbb{Z}_2} = 0$. Obviously, the order parameter $O_{\mathbb{Z}_2 \times \mathbb{Z}_2}$ changes its sign only if a phase transition occurs between D-LD and D-H phases. $O_\mathcal{I}$ behaves similarly to $O_{\mathbb{Z}_2 \times \mathbb{Z}_2}$, i.e., $O_\mathcal{I} = \pm 1$ for the two symmetric phases, and $O_\mathcal{I} = 0$ in the D-AFM phase.

To summarize this subsection, dimerization does not affect the topological properties of the system (2), so that the D-H (D-LD) phase remains a non-trivial (trivial) SPT phase as in the system without dimerization (1).

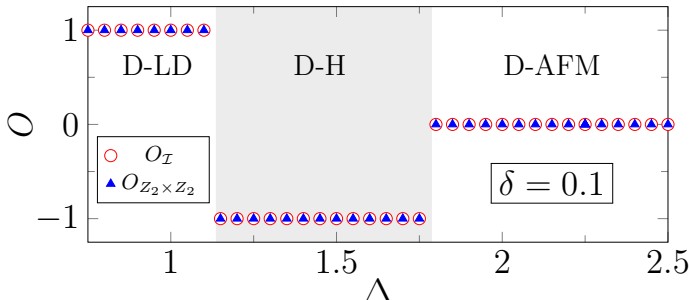

Figure 4: Topological order parameters for inversion symmetry $O_{\mathcal{I}}$ and $\mathbb{Z}_2 \times \mathbb{Z}_2$ spin rotation symmetry $O_{\mathbb{Z}_2 \times \mathbb{Z}_2}$ at $D/J = 1$ and $\delta = 0.1$.

## 5 Relevance to experiments

Let us finally relate our findings with experimental results. There are several realizations of spin-1 bond-alternating chains, such as Ni(C$_9$H$_2$4N$_4$) (NO$_2$)ClO$_4$ [37,38] and [Ni(333-tet)($\mu$-N$_3$)$_n$](ClO$_4$)$_n$ [39–41]. Most remarkably, in the latter material a logarithmic decrease of the susceptibility was observed at low temperature, indicating a vanishing excitation gap [19]. Comparing quantum Monte-Carlo simulations with experimental data suggested that the material is described by a Hamiltonian of the form (2) with $\delta = 0.25$, $\Delta = 1$ and $D/J = 0$. Totsuka *et al.* [42] determined the critical point for $D = 0$ numerically and obtained $\delta_c = 0.25 \pm 0.01$ and $c = 1$, while results by Kitazawa and Nomura [11] suggested that $\delta_c = 0.2598$. Importantly these parameter sets are close to the location of the point where the Gaussian and Ising phase transitions merge [10,11].

In the following, we therefore determine the ground-state phase diagram of the model (2) for $\delta = 0.25$ and reexamine the magnetic susceptibility of the above mentioned nickel compound using the infinite time-evolving block decimation (iTEBD) [35]. Figure 5(a) displays the corresponding phase diagram of the model (2). Although the extent of the Haldane phase is significantly reduced, the Gaussian and Ising transition lines can still be detected numerically. As shown in Fig. 5(b) the experimental data of the magnetic susceptibility for [Ni(333-tet)($\mu$-N$_3$)$_n$](ClO$_4$)$_n$ can be fitted most successfully for $\Delta = 1$ and $D/J = 0.02$, taking the reported small single-ion anisotropy $D/J < 0.1$ [19] into account. On the other hand, the numerical data at the Gaussian transition point for fixed $\Delta = 1$ deviates from experimental ones in the lower-temperature regime. Thus, this nickel compound may be even closer to the Ising transition line than to the c=1 transition line considered so far. It would be interesting to investigate signatures of the Ising QPT experimentally, e.g., by inelastic neutron scattering, where the corresponding dynamical structure factor can be calculated numerically, see Ref. [43].

## 6 Summary and Conclusions

In this work we investigated the ground-state phase diagram and quantum criticality of the dimerized spin-1 $XXZ$ chain with single-ion anisotropy $D$, employing a combination of analytical and numerical techniques. For weak dimerization ($\delta \lesssim 0.26$) and single-ion anisotropy, the symmetry-protected topological Haldane phase survives and the transition between the D-LD and D-AFM phases, which is always of first order in the absence of dimerization, becomes partially continuous. The continuous section of the transition line belongs to the Ising universality class with central charge $c = 1/2$. With increasing the magnitude of $D$, this Ising line terminates at a tricritical Ising point with $c = 7/10$, above which the phase transition becomes first order. A comprehensive description of the phases and phase boundaries can be

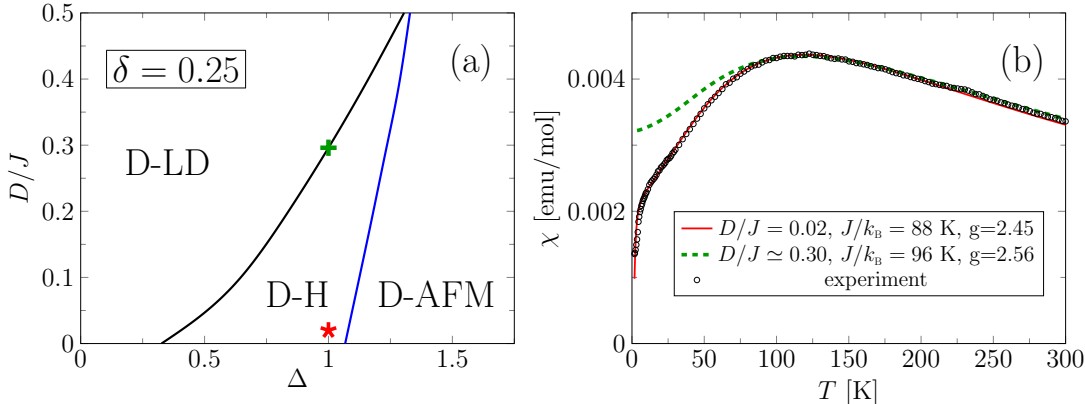

Figure 5: (a) Ground-state phase diagram of the model (2) for $\delta = 0.25$. The red star denotes the parameter set corresponding to the Ni compound [Ni(333-tet)$(\mu$-N$_3)_n$](ClO$_4)_n$, and the green cross gives the Gaussian transition point $[(D/J)_c \simeq 0.296]$ for fixed $\Delta = 1$. (b) Temperature dependence of the magnetic susceptibility of the powdered sample for [Ni(333-tet)$(\mu$-N$_3)_n$](ClO$_4)_n$ (circles) taken from Ref. [19]. The red solid line is the iTEBD data for $\Delta = 1$, $D/J = 0.02$ and $\delta = 0.25$ with $J/k_B = 88$ K and $g = 2.45$. For comparison, we also show the iTEBD result at the Gaussian transition for fixed $\Delta = 1$ (green dashed line).

achieved by a bosonization-based field theory including three Majorana fermions. The field-theory predictions for various correlation functions have been confirmed by numerical iDMRG calculations.

Finally, we have revisited the experimental results for the Ni compound [Ni(333-tet)$(\mu$-N$_3)_n$](ClO$_4)_n$ showing gapless behavior and have demonstrated that the corresponding parameter set might be not only in the vicinity of the Gaussian transition line but also very close to the Ising transition line. Further experimental research for this material, such as neutron scattering, would be desirable.

# Acknowledgements

We thank M. Hagiwara for useful discussions and providing us with their experimental data. The iDMRG simulations were performed using the ITensor library [44].

**Funding information**  This work was supported by the Deutsche Forschungsgemeinschaft (Germany) under Grant No. FE 398/8-1 (FL), by the EPSRC under Grant No. EP/N01930X (FHLE) and the National Science Foundation under Grant No. NSF PHY-1748958 (FHLE). FHLE is grateful to the Erwin Schrödinger International Institute for Mathematics and Physics for hospitality and support during the programme on *Quantum Paths*. TY acknowledges support by a Chiba University SEEDS Fund and YO acknowledges support by a Grant-in-Aid for Scientific Research (Grant No. 17K05530) from JSPS of Japan. HF is grateful to the Los Alamos National Laboratory for hospitality and support.

## A  Low-energy projections of operators

Let us denote the Euclidean action corresponding to the Hamiltonian density (14) by

$$S = S_3 + S_B + S_{\text{int}} \, , \tag{49}$$

where $S_3$ and $S_B$ involve only Ising and bosonic degrees of freedom respectively and $S_{\text{int}}$ describes the interaction between the two sectors. In the regimes where the mass scale associated with $S_3$ is much smaller (larger) than the one associated with $S_B$ and where $S_{\text{int}}$ can be treated as a perturbation, we may integrate out the bosonic (fermionic) degrees of freedom, see e.g. Ref. [33].

### A.1  Integrating out the bosonic degrees of freedom

This case pertains to the transition lines between the D-AFM phase and the D-LD and D-H phases. In these cases the low-energy projection of a general local operator is given by

$$\hat{O}\Big|_{\text{low}} = \int \mathcal{D}\Phi \; e^{-S_B} e^{-S_{\text{int}}} \hat{O} = \langle \hat{O} \rangle_\Phi - \langle S_{\text{int}} \hat{O} \rangle_\Phi + \dots \, , \tag{50}$$

where $\langle \rangle_\Phi$ denotes the average with respect to the bosonic action $S_B$. As we have assumed that the parameter $m$ is positive, we have

$$\langle \sin(\sqrt{4\pi}\Phi) \rangle_\Phi = 0. \tag{51}$$

This implies that the low-energy projection of the dimerization operator is

$$
\begin{aligned}
\hat{D}_j\Big|_{\text{low}} &\sim -\langle S_{\text{int}} \hat{\sigma}^3(x) \sin\left(\sqrt{\pi}\hat{\Phi}(x)\right)\rangle_\Phi + \dots \\
&= -\lambda' \int d\tau dy \; \hat{\sigma}^3(x)\hat{\sigma}^3(y,\tau) \langle \sin\left(\sqrt{\pi}\hat{\Phi}(x,0)\right) \sin\left(\sqrt{\pi}\hat{\Phi}(y,\tau)\right)\rangle_\Phi + \dots \\
&= \langle \hat{d} \rangle + iC\hat{R}_3(x)\hat{L}_3(x) + \dots \, .
\end{aligned}
\tag{52}
$$

In the last line we have used that the expectation value in the bosonic sector decays exponentially in the Euclidean distance $r = \sqrt{(x-y)^2 + v^2\tau^2}$,

$$\langle \sin\left(\sqrt{\pi}\hat{\Phi}(x,0)\right) \sin\left(\sqrt{\pi}\hat{\Phi}(y,\tau)\right)\rangle_\Phi \propto e^{-r/\xi} \, , \tag{53}$$

which in turn allows us to employ the operator product expansion in the Ising sector

$$\hat{\sigma}^3(x)\hat{\sigma}^3(y,\tau) = \left(\frac{a_0}{r}\right)^{\frac{1}{4}} \left[1 - i\pi r \hat{R}_3(x)\hat{L}_3(x)\right] + \dots \, . \tag{54}$$

Finally we have fixed the constant part in the low-energy projection by using that it must give the correct expectation value of the dimerization operator. Similarly we obtain

$$
\begin{aligned}
\hat{M}_j^z\Big|_{\text{low}} &\sim -\lambda' \int d\tau dy \; \hat{\sigma}^3(y,\tau) \langle \partial_x \hat{\Phi}(x,0) \sin\left(\sqrt{\pi}\hat{\Phi}(y,\tau)\right)\rangle_\Phi + \dots \\
&= A\partial_x \hat{\sigma}^3(x) + \dots \, .
\end{aligned}
\tag{55}
$$

The leading contribution to the low-energy projection of $\hat{n}_j^z$ occurs at order $\hat{O}(\lambda')^0$ of our procedure and gives

$$
\begin{aligned}
\hat{n}_j^z\Big|_{\text{low}} &\sim B' \langle \cos\left(\sqrt{\pi}\Phi(\hat{x})\right)\rangle_\Phi \; \hat{\sigma}^3(x) + \dots \\
&= B\hat{\sigma}^3(x) + \dots \, .
\end{aligned}
\tag{56}
$$

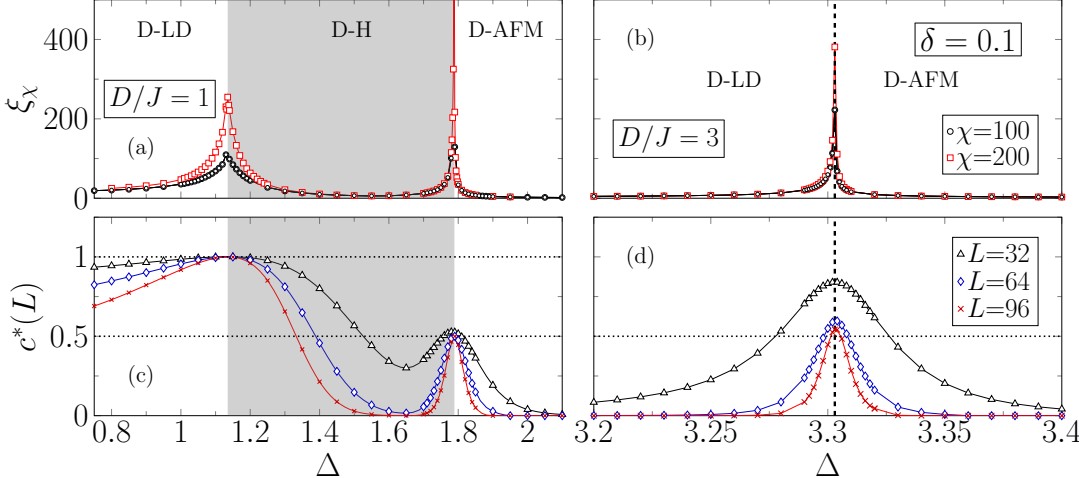

Figure 6: Correlation length $\xi_\chi$ (upper panels) and central charge $c^*(L)$ (lower panels) for fixed $D/J = 1$ (left panels) and 3 (right panels) with $\delta = 0.1$.

### A.2 Integrating out the fermionic degrees of freedom

This case pertains to the transition line between the D-LD and D-H phases. Here we have

$$\hat{O}\Big|_{\text{low}} = \int \mathcal{D}\hat{R}_3 \mathcal{D}\hat{L}_3 \, e^{-S_3 - S_{\text{int}}} \hat{O} = \langle \hat{O} \rangle_3 - \langle S_{\text{int}}\hat{O} \rangle_3 + \dots , \tag{57}$$

where $\langle \rangle_3$ denotes the average with respect to the Majorana action $S_3$. On the transition line we have $m_3 > 0$ which implies

$$\langle \hat{\mu}^3(x) \rangle_3 \neq 0. \tag{58}$$

An immediate consequence of (58) is that

$$\hat{n}_j^x\Big|_{\text{low}} \sim \cos\left(\sqrt{\pi}\hat{\Theta}(x)\right)\langle \hat{\mu}^3(x) \rangle_3 + \dots . \tag{59}$$

The low-energy projections of other operators can be worked out as before

$$
\begin{aligned}
\hat{n}_j^z\Big|_{\text{low}} &\sim -\lambda' B' \int dy\, d\tau \, \langle \hat{\sigma}^3(x,0)\hat{\sigma}^3(y,\tau) \rangle_3 \sin\left(\sqrt{\pi}\hat{\Phi}(x,0)\right)\cos\left(\sqrt{\pi}\hat{\Phi}(y,\tau)\right) \\
&= A_z \sin\left(\sqrt{4\pi}\hat{\Phi}(x)\right) + \dots .
\end{aligned}
\tag{60}
$$

Here we have used that

$$\langle \hat{\sigma}^3(x,0)\hat{\sigma}^3(y,\tau) \rangle_3 \propto e^{-r/\zeta} , \tag{61}$$

which permits us to employ operator product expansions in the bosonic sector. The projection of the dimerization operator is

$$
\begin{aligned}
\hat{D}_j\Big|_{\text{low}} &\sim -\lambda' \int d\tau\, dy \, \langle \hat{\sigma}^3(x)\hat{\sigma}^3(y,\tau) \rangle_3 \sin\left(\sqrt{\pi}\hat{\Phi}(x,0)\right)\sin\left(\sqrt{\pi}\hat{\Phi}(y,\tau)\right) + \dots \\
&= \langle \hat{d} \rangle + D \cos\left(\sqrt{4\pi}\hat{\Phi}\right) + \dots .
\end{aligned}
\tag{62}
$$

## B Determination of phase boundaries

In this section, we explain how the QPT points and their universality classes are determined within the (i)DMRG method. Since the QPTs are the only points in the considered parameter

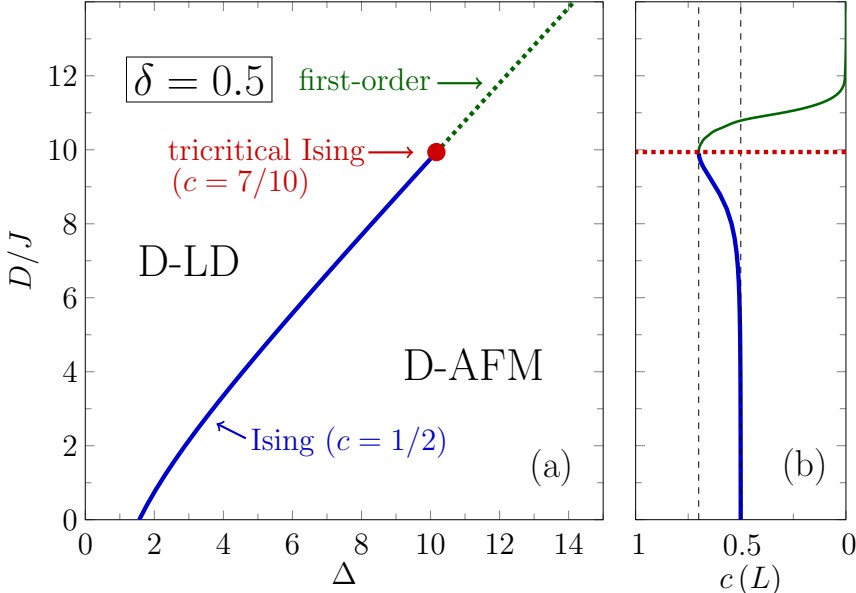

Figure 7: (a) Phase diagram of the model (2) for $\delta = 0.5$. D-LD⇋D-AFM phase boundary of the continuous Ising transition terminates at a tricritical Ising point. Beyond this point, the QPT becomes first order. (b) Central charge $c(L)$ on the D-LD⇋D-AFM phase boundaries obtained numerically for $L = 128$ and periodic boundary conditions.

region where the system becomes critical, they are easily obtained by simulating the correlation length $\xi_\chi$, as demonstrated in Figs. 6(a) and (b) for $\delta = 0.1$ with fixed $D/J = 1$ and 3, respectively. The divergence of the physical correlation length at a QPT is reflected by a pronounced peak of $\xi_\chi$ whose height increases with the bond dimension $\chi$. From the peak positions for large enough $\chi$, we pinpoint the phase transition with an accuracy of at least three digits. For $D/J = 1$ the transitions occur at $\Delta_{c1} \simeq 1.135$ and $\Delta_{c2} \simeq 1.789$ [see Fig. 6(a)], while there is only one Ising transition at $\Delta_c \simeq 3.303$ [see Fig. 6(b)].

The central charge $c^*(L)$ calculated by DMRG also exhibits a peak structure around the critical points [see Figs. 6(c) and (d)]. These peaks become more distinct with increasing system size $L$. From the heights of the peaks at large $L$, we obtain the central charges $c = 1$ and $c = 1/2$, which are consistent with Gaussian- and Ising-type transitions, respectively. Moreover, the positions of the peaks agree with the QPT points estimated from the correlation length.

## C    Ground-state phase diagram for strong dimerization

With increasing dimerization the D-H phase is reduced, and it disappears for $\delta \gtrsim 0.26$ [11] if we limit ourselves to the parameter region $J > 0$ and $\delta > 0$. Figure 7(a) for $\delta = 0.5$ demonstrates such a situation consisting of only D-LD and D-AFM phases, separated by continuous and first-order transition lines. At the meeting of these transition lines the numerically obtained central charge indicates $c = 7/10$ [Fig. 7(b)], suggesting that this point belongs to the tricritical Ising universality class.

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
