# Peer review of "Exotic criticality in the dimerized spin-1 $XXZ$ chain with single-ion anisotropy"

_SciPost Physics, doi:SciPost Phys. 5, 059 (2018)_

## Round 1 · Referee Report · Anonymous · 2018-10-20

Strengths

1. The paper provides a comprehensive study of the phase diagram of the spin-1 XXZ chain with single-ion anisotropy.
2. The numerical results are corroborated by an extensive field-theory analysis.
3. The presentation is very clear and the obtained results are of interest both as a theoretical contribution to quantum criticality as well as for experimental studies of certain spin chain compounds.

Weaknesses

1. There is almost no information given about the accuracy of the DMRG calculations (e.g. discarded weights in general, error bars for the phase boundaries and tricritical points).

Report

The authors present a comprehensive study of the phase diagram of the spin-1 XXZ chain with dimerization and single-ion anisotropy. Numerical results obtained by DMRG are analyzed in terms of a field theory containing both an Ising and a bosonic sector.

The presentation is very clear and the various crosschecks between DMRG and field theory leave no doubt that the phase diagram obtained is qualitatively correct. Overall, this is a very nice work and an important contribution to the field.

The only weakness in my view is the lack of details with regard to the numerical calculations, see below.

Requested changes

The phase diagram shown in Fig. 1 consists of a D-LD and a D-AFM phase with a small D-H phase 'squeezed' in between these two at small Delta and D. This is qualitatively reminiscent of phase diagrams of extended Hubbard models (the phases and critical lines are, of course, different) which have been extensively studied using DMRG and field theories more than 10 years ago. A quantitative analysis of the latter phase diagrams by DMRG has turned out to be quite difficult. Determining the exact position of the tricritical point, for example, has required a careful extrapolation in the truncation error and system size.

In this light I would like to ask the authors to give more details about their numerical results including truncation errors (have the data been extrapolated?), and error bars for the phase transition lines and the tricritical point. For the experimentally potentially relevant case shown in Fig. 5, in particular: How certain can the authors be that the D-H phase and two separate phase transitions still exist for this value of dimerization?

  • validity: high
  • significance: high
  • originality: good
  • clarity: top
  • formatting: perfect
  • grammar: perfect

Author:  Satoshi Ejima  on 2018-11-19  [id 344]

(in reply to Report 1 on 2018-10-20)
Category:
answer to question

We would like to thank the Referee for Her/His careful reading of the manuscript and the helpful comments.

In the revised version we comment on the accuracy level of our numerical scheme, discussing the discarded weights in the beginning of Section 2 and the precision of the tricritical point in the caption of Fig.1 (actually the error bar now is significant smaller than the symbol size). Estimating the error bar of the tricritical point we corrected the given value of the tricritical point somewhat compared to the previous version.

Moreover, we added a new appendix (Appendix B) in order to demonstrate the high accuracy of the transition points obtained by the infinite DMRG (iDMRG) technique. In our model, all three phases are gapped, and the system becomes critical only on the continuous transition lines. Therefore the correlation length can be used to pinpoint the transition point by determining the correlation length for various $\Delta$ at fixed $D/J$ as shown in Fig.6. Thereby a finite-size scaling is, however, not necessary since we use the iDMRG technique, which provides physical quantities directly in the thermodynamic limit.

Differently from our model, in the half-filled extended Hubbard model with nearest-neighbor Coulomb interaction the spin degrees of freedom are gapless in the spin-density-wave (SDW) phase, which makes it difficult to determine the phase boundaries between SDW and the so-called bond-order-wave (BOW) phases. Hence the correlation length obtained by iDMRG cannot be used to determine, e.g., the SDW-BOW transition point.

The phase diagram for $\delta=0.25$ (Fig.5) can be produced in the same way. In the absence of the single-ion anisotropy $D$, the D-H phase disappears at about $\delta\simeq 0.26$ from the ground-state phase diagram in the $\delta$-$\Delta$ plane, see Ref. [10,11]. Therefore, our observation of a D-H phase in the $D/J$-$\Delta$ plane for $\delta=0.25$ corroborates previous results.

In addition, we added an explanation for the topological order parameter in the presence of dimerization shortly after Eq. (48). See also reply to Report 2 for further modifications in the revised manuscript.

---

## Round 1 · Referee Report · Anonymous · 2018-11-9

Strengths

1) The authors present a combined field-theoretical and numerical study of the
dimerized spin-1 XXZ chain with single-ion anisotropy. In the
field-theoretical part the authors derive the scaling dimensions of the
critical lines, Ising and Gaussian, where for the Gaussian line actually a TLL
behaviour is found. In any case, these scaling forms are consistent with the
DMRG calculations.

It seems that the DMRG calculations are slightly stronger as they allow to
detect the tricritical Ising point with central charge c=7/10.

2) The presentation is very good.

Weaknesses

1) It is not clear to me why the authors call the transition line with TLL
exponents Gaussian? Is it because <m_0m_r>~1/r^2? Or do they simply call a
c=1 transitian line Gaussian since the field theory may correspond to a
(compactified, orbifolded) free boson as used in bosonization?

2) On p.5 the authors comment

"Our main working assumption is that the field theory (4) remains a good
description of the low-energy degrees of freedom in the vicinity of the
various phase transition lines in the microscopic model even far away in
parameter space from the Takhtajan–Babujian point."

I think, with regard to the inclusion of dynamical dimerization, the
Takhtajan-Babujian point is rather special as it is on the border of the
Haldane phase and the spontaneously dimerized phase: including the biquadratic
term in (4) yields the model dimerized for alpha<0 even without dynamical
symmetry breaking. In this regard, approach 1 for setting up a field theory is
distinct from the alternatives 2 and 3. Maybe the authors can comment on
this. In any case, the results (a posteriori at least) somehow justify the
approach.

3) I have some concern regarding the application of the model to the Ni compound
[Ni(333-tet)(μ-N3 )n ](ClO4 )n. I find it astonishing / unlikely that the
coupling parameters should fall in a 2d phase diagram extremely close to /
onto a critical line. Furthermore, the low T-behaviour of the susceptibility
(Fig.5b) reminds me of the isotropic spin-1/2 chain with logarithmic
corrections. Therefore, locating the coupling parameters onto an Ising (!)
line looks very daring. Do the authors expect paramagnetic ions be responsible
for the steep drop?

Report

I recommend publication after the authors have considered the formulated questions and the suggested amendments.

Requested changes

1) Fig.1: Please explain the meaning of <sigma^3> in the caption.

2) p.10:
Let us now the explore topological properties ...
->
Let us now explore topological properties ...

3) p.13
Is the following sentence grammatically correct?

"In the regimes where the mass scale associated with S3 is much smaller
(larger) than the one associated with SB and Sint can be treated as a
perturbation, we may integrate out the bosonic (fermionic) degrees of freedom,
see e.g. Ref. [33]."

  • validity: top
  • significance: top
  • originality: high
  • clarity: high
  • formatting: perfect
  • grammar: perfect

Author:  Satoshi Ejima  on 2018-11-19  [id 345]

(in reply to Report 2 on 2018-11-09)

We would like to thank the Referee for Her/His careful reading of the manuscript and the helpful comments.
In the following, we address the three points of “Weaknesses” and three points of “Requested changes” raised by the Referee.
(See also reply to Report 1 for further modifications in the revised manuscript.)

Weaknesses 1:
Our usage of “Gaussian model” for the free (compactified) relativistic boson in 1+1 dimensions is standard in the field, see e.g. the textbook by Gogolin, Nersesyan and Tsvelik on bosonization.

Weaknesses 2:
As usual field theoretic RG is defined in the vicinity of a critical point. In our case this is the Takhtajan-Babujian (TB) point and we may describe any (small) perturbation away from it as we know the exact operator content at the critical point. So the applicability of our field theory description in the vicinity of the TB point is guaranteed, irrespective of which perturbations (dimerization, biquadratic interaction etc.) we apply. The applicability of the same field theory description, with appropriately adjusted coupling constants at the cutoff scale, is as we clearly state an assumption. This assumption is verified a posteriori by comparing the field theory descriptions to DMRG computations. As we noted in our manuscript, using Schulz's bosonization prescription leads to equivalent results.

Weaknesses 3:
The steep drop of the susceptibility suggests that this Ni-compound lies at (or close to) the Gaussian or Ising transition point, where the system becomes gapless. In the case of the Gaussian transition with central charge $c=1$, the behavior of the susceptibility might be similar to the spin-1/2 Heisenberg chain as suggested by the Referee. Note that the logarithmic correction in the spin-1/2 Heisenberg chain becomes only visible in the very low-temperature regime, see, e.g., Eur. Phys. J. B 5, 677 (1998).

In Fig.5(b) we have demonstrated that the experimental data of Ref.[19] can be reproduced by the modern numerical technique (infinite TEBD) with the same parameter set $\delta=0.25$ as used in Ref. [19] (note that the very small single-ion anisotropy $D/J=0.02$ was also reported in Ref.[19]). Figure 5(a) shows the phase diagram in the $D/J$-$\Delta$ plane, where the corresponding parameter set is denoted by the star symbol. If the Ni-compound would have been described by a larger value of $D/J$, it might be located on the $c=1$ line (“plus” symbol), however, the steep drop of the susceptibility in the low-temperature regime doesn’t show up [compare the dashed line in Fig.5(b)]. Here a sharp drop of the susceptibility will appear at extremely low temperatures at most, just as for the spin-1/2 Heisenberg chain. Therefore we are led to the conclusion that a reasonably well fit of the experimental data indicates that the Ni compound is parametrized close to the Ising transition point.

Requested changes 1:
We added a sentence with respect to $\langle\sigma^3\rangle$ in the caption of Fig.1:
“$\langle \sigma^3\rangle$ denotes third Ising order parameter, determining the Ising QPT between the D-H or D-LD phase and the D-AFM phase.”

Requested changes 2:
We have modified the corresponding sentence in the revised version.

Requested changes 3:
We have modified the sentence as follows (added “where”):
In the regimes where the mass scale associated with $S_3$ is much smaller (larger) than the one associated with $S_B$ and “where" $S_{\rm int}$ can be treated as a perturbation, we may integrate out the bosonic (fermionic) degrees of freedom, see e.g. Ref. [33].

Anonymous on 2018-11-21  [id 349]

(in reply to Satoshi Ejima on 2018-11-19 [id 345])
Category:
answer to question

I thank the authors for their exhaustive answers to my questions. The presented scientific work is a strong analysis of the phase diagram and the thermodynamics of dimerized spin-1 Heisenberg chains. I strongly recommend this manuscript for publication in SciPost.

---

## Round 2 · Referee Report · Anonymous (Referee 1) · 2018-11-21

Report

I am satisfied with the changes in response to my report. I recommend publication of the manuscript in its current form.

---

## Editorial Decision

published